# Expression and Function of ABC Transporters in Human Alveolar Epithelial Cells

**DOI:** 10.3390/biom12091260

**Published:** 2022-09-07

**Authors:** Rossana Visigalli, Bianca Maria Rotoli, Francesca Ferrari, Maria Di Lascia, Benedetta Riccardi, Paola Puccini, Valeria Dall’Asta, Amelia Barilli

**Affiliations:** 1Laboratory of General Pathology, Department of Medicine and Surgery, University of Parma, 43125 Parma, Italy; 2Preclinical Pharmacokinetics, Biochemistry & Metabolism Department, Chiesi Farmaceutici, 43122 Parma, Italy

**Keywords:** alveolar epithelium, ATP-binding cassette transporters, BCRP, MDR1, MRP1

## Abstract

ATP-binding cassette (ABC) transporters are a large superfamily of membrane transporters that facilitate the translocation of different substrates. While ABC transporters are clearly expressed in various tumor cells where they can play a role in drug extrusion, the presence of these transporters in normal lung tissues is still controversial. Here, we performed an analysis of ABC transporters in EpiAlveolar^TM^, a recently developed model of human alveoli, by defining the expression and activity of MDR1, BCRP, and MRPs. Immortalized primary epithelial cells hAELVi (human alveolar epithelial lentivirus-immortalized cells) were employed for comparison. Our data underline a close homology between these two models, where none of the ABC transporters here studied are expressed on the apical membrane and only MRP1 is clearly detectable and functional at the basolateral side. According to these findings, we can conclude that other thus-far-unidentified transporter/s involved in drug efflux from alveolar epithelium deserve investigations.

## 1. Introduction

ATP-binding cassette (ABC) transporters are a ubiquitous superfamily of integral membrane proteins responsible for the ATP-dependent transport of a wide variety of substrates [1,2,3,4]. These active efflux transporters are expressed in various tissues such as the liver, intestine, kidney, and brain, where they limit the entry of infectious pathogens, particulate matter, and other xenobiotics [5]; however, the broad substrate specificity of ABC transporters also allows their interaction with pharmaceutical compounds intended for medical therapy, and, thus ascribes them a significant role in determining the efficacy and toxicity profile of many drugs [6]. In this context, multidrug resistance 1 (*ABCB1*/MDR1, previously known as P-glycoprotein), multidrug resistance-associated proteins (*ABCC*/MRPs), and breast cancer resistance protein (*ABCG2*/BCRP) are considered the most relevant members as far as the transport of pharmacological compounds is concerned [7,8,9]. These efflux pumps gain particular relevance in the lung, where their activity may be pivotal in driving the availability of inhaled drugs [10,11].

The expression of ABC transporters in human lungs is to date still quite controversial: they have been well-studied in lung cancer, where their upregulation confers multidrug resistance to tumor cells, resulting in a poor outcome for patients [12,13]; conversely, reports about their level of expression, localization, and activity in the normal tissue are still conflicting [2,10,11]. However, the increasing development of drug delivery systems is raising the need of a precise definition of ABC expression and activity in pulmonary cells, as well as of reliable in vitro cell models for the study of drug disposition in the lung, and particularly in the alveolar epithelium, where the absorption of inhaled drugs takes place.

Two types of alveolar epithelial cells have been described to date: ATI, which usually occupies 95% or the alveolar surface area and forms the epithelial component of the thin air–blood barrier for the transport of gases [14], and ATII, cuboidal alveolar cells that serves as alveolar stem cells able to divide and differentiate into ATI cells, to replace and repair the epithelium in case of lung injury [15]. 

Although it is undeniable that freshly isolated human primary cells would be the most realistic model of the situation in vivo, it is also true that the scarce availability and the difficulty of maintaining them in culture limit their use on a large scale. For this reason, two human airway epithelial cell lines, A549 and NCI-H441, have been extensively employed over the years as models of the alveolar epithelium for biopharmaceutical research [16,17,18,19,20]. Both cells, however, endowed with an ATII-like phenotype, were isolated from a human pulmonary adenocarcinoma, and hence can present differences with respect to the physiological epithelium in vivo. Alternatively, an appreciated model for in vitro transport studies across the alveolar epithelium is represented by hAELVi (human alveolar epithelial lentivirus-immortalized cells), a cell line of immortalized primary human alveolar cells that, by expressing functional junctions, form tight, impermeable monolayers [21]. More recently, a commercial cell system closely reproducing human alveoli in vitro has been developed, the EpiAlveolar^TM^. This tissue is an air–liquid interface coculture model of the air–blood barrier that consists of human primary alveolar epithelial cells, pulmonary endothelial cells, and fibroblasts. Alveolar epithelial cells, both ATI- and ATII-like, are cultured on the top surface of microporous membrane filters, with endothelial cells localized on the underside of the membrane; fibroblasts are layered on the membrane under alveolar cells. Thanks to this structure, this culture system reproduces at best the features of the alveolar epithelium in vitro. 

In this context, the aim of the present study is to verify the reliability of EpiAlveolar^TM^ as a truthful model of alveoli in vivo by defining the expression and activity of *ABCB1*/MDR1, *ABCCs*/MRPs, and *ABCG2*/BCRP with respect to hAELVi, employed for comparison as normal alveolar epithelial cells. 

## 2. Materials and Methods

### 2.1. Cell Cultures 

EpiAlveolar™ (ALV-100-FT-PE12), an in vitro organotypic 3D model of the human alveolar tissue, was obtained from MatTek Corporation (Ashland, MA, USA) and grown according to the manufacturer’s instructions. 

Human alveolar epithelial lentivirus-immortalized cells (hAELVi), purchased from InSCREENeX GmbH (Braunschweig, Germany), were routinely cultured in huAEC Medium (InSCREENeX GmbH) added with 1% penicillin/streptomycin. For the experiments, hAELVi monolayers were grown under air–liquid interface (ALI) conditions; to this end, cells were seeded onto cell culture inserts (0.33 cm^2^, 0.4 µm pore size; Falcon®, VWR International, Milano, Italy) at the density of 10^5^ cells/insert; the medium in the apical chamber was removed 48 h after seeding, while basolateral medium was replaced with hAELVi FasTEER medium (InSCREENeX GmbH), which was renewed every other day. The monolayers were allowed to differentiate under ALI conditions for 21 days. All hAELVi culture devices were precoated with huAEC coating solution (InSCREENeX GmbH) before cell seeding.

A549 and Calu-3 cell lines, obtained from American Type Culture Collection (ATCC, Manassas, VA, USA), were employed as positive controls in expression studies. A549 cells were routinely grown on plasticware in high-glucose DMEM, added with 10% fetal bovine serum (FBS; Merck, Milano, Italy), while Calu-3 cells were cultured in Eagle’s Minimum Essential Medium (EMEM) supplemented with 10% FBS and 1 mM sodium pyruvate. Cells, employed between passages 25-30, were routinely cultured under physiological conditions (37.5°C, 5% CO_2_, 95% humidity) in 10 cm-diameter dishes. For the experiments, Calu-3 monolayers were grown at ALI conditions for 21 days; to this end, 10^5^ cells were seeded onto cell culture inserts and the apical medium was removed 24 h after seeding, while basolateral medium was renewed every other day.

### 2.2. Transepithelial Electrical Resistance (TEER) and Cellular Permeability

The integrity of cell monolayers was verified by measuring transepithelial electrical resistance (TEER) with an epithelial voltmeter (EVOM, World Precision Instruments). Commonly TEER values > 300 Ω/cm^2^ are accepted for confluent tight monolayers [22,23,24]; we employed cell culture when they reached values > 500 Ω/cm^2^.

The paracellular and transcellular permeability of cell monolayers to solutes was assessed by monitoring the apical-to-basolateral fluxes of ^14^C-mannitol and ^3^H-propranolol, respectively. In detail, cell monolayers were washed and incubated for 30 min in Hank’s Balanced Salt Solution (HBSS, pH 7.4, 37 °C); HBSS containing ^14^C-mannitol (1 µCi/mL, corresponding to 20 µM) or ^3^H-propranolol (3 µCi/mL; 10 µM) was then added to the apical side. Aliquots of medium were collected from the basolateral compartment after 0, 30, 60, and 120 min and replaced with fresh HBSS; radioactivity in each sample was measured with a MicroBeta^2^ liquid scintillation spectrometer (Perkin Elmer, Italy) and employed to calculate the apparent permeability coefficient (P*_app_*). Mannitol P*_app_* values < 5 × 10^−6^ cm/s were accepted as an index of tight monolayers [25,26].

### 2.3. Bidirectional Transport Studies

The activity of ABC transporters was determined by measuring the apical-to-basolateral (AB) and basolateral-to-apical (BA) fluxes of specific substrates. To this aim, cell monolayers were washed twice in HBSS and equilibrated for 30 min in the same solution (pH 7.4, 37 °C). Either the apical or the basolateral compartment (donor chamber) was then incubated in HBSS added with the labeled substrates Rhodamine123 (1µM) for MDR1, ^3^H-Estrone-3-sulfate (0,5 µM, 3 µCi/ml) for MRPs or Bodipy™ FL Prazosin (3µM) for BCRP; the inhibitors PSC833, MK-571, Ko143, and febuxostat were employed when indicated. After 0, 30, 60, and 120 min, aliquots of the solution were collected from the opposite chamber (receiver chamber) and replaced with the same volume of HBSS. The fluxes were determined by reading fluorescence with an EnSpire® Multimode Plate Reader (Perkin Elmer), or radioactivity with MicroBeta^2^ liquid scintillation spectrometer. The apparent permeability coefficient (P*_app_*) was calculated from the data obtained. TEER was measured before and after the experiment to verify the integrity of cell monolayers; values were not modified by the experimental procedures.

### 2.4. Calculation of P_app_

The apparent permeability coefficient (P*_app_*) of the tracer molecules was calculated according to the equation:P*_app_* = (dQ⁄dt)⁄(A×C_0_×60)
where dQ/dt is the transport rate of the tracer, A is the filter surface area, C_0_ is the initial concentration of the tracer in the donor chamber, and 60 is the conversion from minutes to seconds.

Efflux ratio (ER) was calculated as the ratio between P*_app_* measured for BA and AB fluxes, while uptake ratio (UR) was the ratio between P*_app_* measured for AB and BA fluxes.

### 2.5. Absolute Quantification of mRNA Expression

RT-qPCR was employed for the analysis of mRNA, as previously described [27]. Briefly, 1µg of total RNA was reverse-transcribed and 20 ng of cDNA underwent qPCR on a StepOnePlus Real-Time PCR System (Thermo Fisher Scientific, Monza, Italy). The amount of *ABCB1* (NM_000927.4), *ABCC1* (NM_004996.4), *ABCC2* (NM_000392.5), and *ABCG2* (NM_001257386.2), and that of the reference gene RPL15 (Ribosomal Protein Like 15; NM_001253379.2), were measured employing specific TaqMan® Gene Expression Assays (Thermo Fisher Scientific; Cat# Hs00184500_m1, Hs01561512_m1, Hs00960488_m1, Hs00960489_m1, and Hs03855120_g1, respectively). The absolute quantification of mRNA molecules was performed as described previously [28].

### 2.6. Western Blot Analysis

Protein expression was determined in cell lysates in LDS sample buffer (Thermo Fisher Scientific, Monza, Italy) and Western blot analysis was performed as described [29]. A total of 20 µg of proteins was separated on Bolt™ 4-12% Bis-Tris mini protein gel (Thermo Fisher Scientific) and electrophoretically transferred to PVDF membranes (Immobilione-P membrane, Merck). Membranes were incubated for 1 h at RT in blocking solution (TBST: 50 mM Tris-HCl pH 7.5, 150 mM NaCl; 0.1% Tween containing 5% non-fat dried milk) then incubated overnight at 4 °C in TBST added with 5% bovine serum albumin (BSA) and anti-MDR1 (E1Y7B, #13342), anti-MRP1 (D7O8N, #14685), anti-MRP2 (D9F9E, #12559), or anti-BCRP (D5V2K, #42078) rabbit polyclonal antibody (1:1000; Cell Signaling Technology, Euroclone, Milano, Italy). Vinculin, visualized by a mouse monoclonal antibody (1:2000; SAB4200080, Merck), was employed as internal loading control. Membranes were then incubated for 1 h in blocking solution containing HRP-conjugated secondary antibodies (anti-rabbit #7074 or anti-mouse (#7076) IgG, (1:10,000, Cell Signaling Technology). Immunoreactivity was detected by employing SuperSignal™ West Pico Plus Chemiluminescent HRP Substrate; and blot images, taken with iBright FL1500 Imaging System, were analyzed with iBright Analysis Software (Thermo Fisher Scientific).

### 2.7. Immunocytochemistry

For immunocytochemistry, cell monolayers, grown on cell culture inserts, were rinsed in phosphate buffered saline (PBS) and incubated for 15 min at RT in 3.7% paraformaldehyde. After three washes in PBS, cells were incubated for 1 h at 37 °C in PBS containing 10% BSA and 0.3% Triton-X-100, so as to permeabilize cell membranes and block unspecific binding sites, then maintained overnight at 4 °C in PBS added with 3% BSA and anti-MRP1 polyclonal antibody (1:100, Cell Signaling Technology). At the end, monolayers were incubated for 45 min with anti-rabbit IgG Alexa Fluor® 488 conjugate (1:400, Cell Signaling Technology); Propidium Iodide (PI/RNase Staining Solution, Cell Signaling Technology) was added to stain nuclei. At the end, permeable filters, cut from inserts, were layered on glass slides and mounted with fluorescence mounting medium (FluorSave Reagent, Calbiochem, Merck). Immunostained cells were visualized with the confocal system Stellaris 5 (Leica Microsystems, Wetzlar, Germany) using a 40x (1.3 NA) oil objective. 

### 2.8. Statistical Analysis

Statistical analysis was performed using GraphPad Prism® 9 (GraphPad Software, San Diego, CA, USA). All data were analyzed with a two-tailed Student’s *t*-test. *p* values < 0.05 were considered statistically significant.

### 2.9. Materials

^3^H-Estrone-3-sulfate was purchased from Perkin Elmer and Bodipy™ FL Prazosin from Thermo Fisher Scientific. Unless otherwise specified, all chemical and reagents were from Merck.

## 3. Results

The barrier properties of hAELVi and EpiAlveolar™ cell cultures were preliminary addressed by measuring the transepithelial electrical resistance (TEER) of the monolayers, as well as their para- and transcellular permeability to fluxes of mannitol and propranolol, respectively. As shown in Figure 1, the two cell models displayed a comparable permeability to propranolol. Conversely, a slight not significant difference was observed when measuring the paracellular permeability of the monolayers, with EpiAlveolar™ being more permeable to mannitol than hAELVi; consistently, TEER values were significantly higher in hAELVi cultures than in EpiAlveolar™, thus further supporting the weaker tightness of these latter cells. 

The comparison between the two cellular models was then conducted by addressing the activity and expression of three well-known ABC transporters. To this end, the activity of MDR1 was first determined by measuring the bidirectional fluxes of the fluorescent substrate Rhodamine123, both in the absence and in the presence of MDR1 inhibitor PSC833 [30,31] (Figure 2A). Results obtained indicated that the P*_app_* values were comparable for AB and BA fluxes, thus excluding a directionality of the transport, as demonstrated by comparable values of efflux ratio (ER) and uptake ratio (UR) in EpiAlveolar™ and hAELVi; moreover, since P*_app_* was not modified by the addition of the inhibitor, our findings indicate an at least negligible activity of the transporter in both cell models. Consistently, the analysis of the expression of the transporter demonstrated that the number of mRNA molecules coding for MDR1 was very low—about 80-fold lower with respect to Calu-3 maintained under ALI conditions for 21 d [32], employed as positive control. Accordingly, MDR1 protein was undetectable in both EpiAlveolar™ and hAELVi, as well as in A549 cells (Figure 2B). 

Similarly, the activity of *ABCG2*/BCRP transporter was evaluated by measuring the transcellular fluxes of Bodipy™ FL Prazosin, employed as a tracer. As shown in Figure 3A, P*_app_* values for AB and BA fluxes were almost identical, reflecting comparable values of ER and UR in EpiAlveolar™ and hAELVi. Moreover, no inhibition was observed when either febuxostat or Ko143, inhibitors of BCRP [33,34], were employed, thus excluding any activity of the transporter in both cell models. In line with this finding, the mRNA for BCRP was tenfold lower than in A549 and the protein was completely undetectable, hence confirming the absence of the transporter in both EpiAlveolar™ and hAELVi monolayers.

Lastly, the activity of *ABCCs*/MRPs was evaluated by monitoring the fluxes of ^3^H-estrone-3-sulfate, both in the absence or in the presence of inhibitor MK571 [35,36] (Figure 4A). In EpiAlveolar, P*_app_* values were higher for AB than for BA fluxes, yielding values of UR higher than ER, indicating an MRP-like activity at the basolateral side. Consistently, the inhibitor proved effective in inhibiting AB but not BA fluxes in both EpiAlveolar™ and hAELVi monolayers, confirming the operativity of an MRP transporter on the basolateral side. A molecular analysis was then performed to monitor the expression of *ABCC1*/MRP1 and *ABCC2*/MRP2. The mRNA for MRP2 was very scarce and the protein completely undetectable in both cell models; conversely, MRP1 appeared readily detectable both as mRNA and protein, although less abundant than in A549, employed as positive control (Figure 4B). In line with transport data, the images obtained with confocal microscopy showed a continuous staining of MRP1 around cell borders on the plasma membrane of both hAELVi and EpiAlveolar™, confirming the basolateral localization of this transporter.

## 4. Discussion

Given the relevance of ABC transporters in modulating the availability of inhaled drugs in the lungs, we here addressed the expression and activity of *ABCB1*/MDR1, *ABCCs*/MRPs, and *ABCG2*/BCRP in a 3D system recently developed to reproduce human alveoli in vitro, the EpiAlveolar™. According to our findings, neither MDR1 nor BCRP were found in these cells, while an ABCC-mediated activity was observed; consistently, MRP1 was readily detectable at both the gene and protein level, with a clear-cut localization at the basolateral side of the monolayers. Results completely overlapping were obtained in immortalized primary alveolar epithelial cells, hAELVi.

The lack of MDR1 in alveolar cells deserves particular attention given the fundamental role of this transporter in drug extrusion. Thus far, only a few contributions have addressed this issue, and the presence of this transporter in the alveoli is still a matter of debate. A modest positivity to MDR1 has been observed in A549 tumor cells [37,38], where we did not detect any expression of this transporter. As far as normal cells are concerned, immunostaining for MDR1 was observed across the entire alveolar epithelial surface of normal distal lung tissues, although only faint bands for the mRNA and protein of the transporter have been detected in epithelial cells isolated from the same tissue [39]. A minimal expression of MDR1 has also been reported in paraffin-embedded lung sections obtained from human healthy subjects [40], while we recently demonstrated no expression of this transporter neither in EpiAirways™, a model of normal human bronchial cells, nor in specimens of normal human bronchi [32]. Although the issue deserves to be further addressed, these findings suggest an almost negligible presence of MDR1 in epithelial cells of peripheral lungs.

In our hands, BCRP appears active in neither EpiAlveolar™ nor in hAELVi, where the mRNA coding for the transporter is barely appreciable and the corresponding protein undetectable; the protein is instead readily detectable in A549 tumor cells. Park et al. stated that healthy human lungs minimally express BCRP [40]. Conversely. the transporter has been detected in NCI-H441 cell line, as well as in ATII pneumocytes, but not in transdifferentiated ATI-like cells [41]. These findings, by highlighting the differences between normal and neoplastic airway epithelial cells, provide evidence that A549 and NCI-H441 are not a proper in vitro model for the study of BCRP in human distal lung.

On the contrary, an ABCC-mediated activity was detected at the basolateral side of both EpiAlveolar™ and hAELVi. Among the nine members of the ABCC family (MRP1-9) identified thus far, MRP1 and 2 are the most-studied. The lack of expression of MRP2 against a clear-cut expression of MRP1 ascribes a role to at least this latter transporter in the observed fluxes of estrone-3-sulfate. Accordingly, the immunocytochemistry analysis confirmed the localization of MRP1 on the basolateral membrane. Similar findings have been obtained in the NCI-H441 cell line, as well as in human ATI and ATII primary cells, with the expression of the transporter increasing upon differentiation from ATII to ATI-like phenotype [42]. Moreover, we recently showed a similar activity and localization in EpiAirways™ and Calu-3 bronchial cells [32]. 

In conclusion, our results demonstrate that none of the ABC transporters analyzed are present and operative at the apical membrane of EpiAlveolar™ and hAELVi, while only MRP1 is detectable at the basolateral side. In a recent contribution we demonstrated that not even organic cation transporters (OCTs/OCTNs) are present at the apical side of EpiAlveolar™, where only ATB^0,+^ transporter is detectable [29]. However, since the activity of this latter transporter is known to mediate the influx of substrates, we can hypothesize that other thus-far-unidentified transporter/s are responsible for drug efflux from alveolar epithelial cells.

## Figures and Tables

**Figure 1 biomolecules-12-01260-f001:**
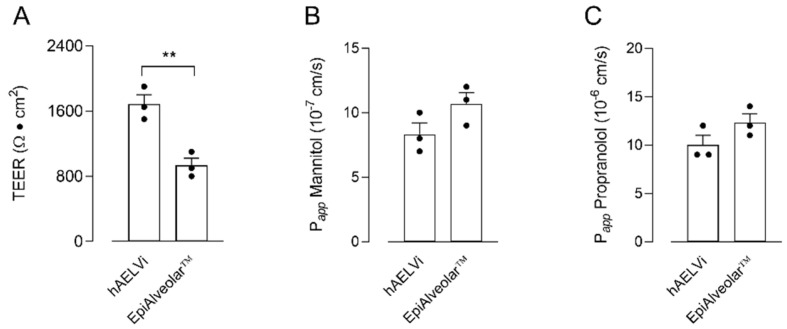
Transepithelial electrical resistance (TEER) and cellular permeability (P*_app_*) in EpiAlveolar™ and hAELVi. TEER values (panel (**A**)) and P*_app_* of mannitol (panel (**B**)) and propranolol (panel (**C**)) were measured in cells cultured under air–liquid interface (ALI) conditions, as described in Materials and Methods. Bars are the mean ± SEM of 3 independent determinations (single dots). ** *p* < 0.01 with a two-tailed Student’s *t*-test.

**Figure 2 biomolecules-12-01260-f002:**
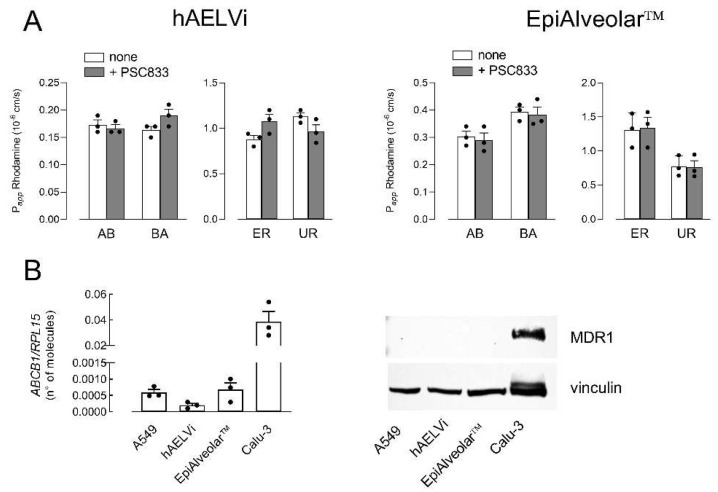
Expression and activity of MDR1. Panel (**A**): The apical-to-basolateral (AB) and basolateral-to-apical (BA) fluxes of 1 µM Rhodamine123 were measured both in the absence (none) and in the presence of 10 µM PSC833, as indicated. Data obtained were employed to calculate the efflux ratio (ER) and uptake ratio (UR), as defined in Materials and Methods. Bars are the mean ± SEM of 3 independent determinations (single dots). Panel (**B**): The expression of MDR1 gene and protein were performed with RT-qPCR and Western blot analyses, respectively, as described in Materials and Methods. Data of mRNA expression are the mean ± SEM of 3 independent determinations (single dots). A representative Western blot is shown that, repeated twice, gave comparable results.

**Figure 3 biomolecules-12-01260-f003:**
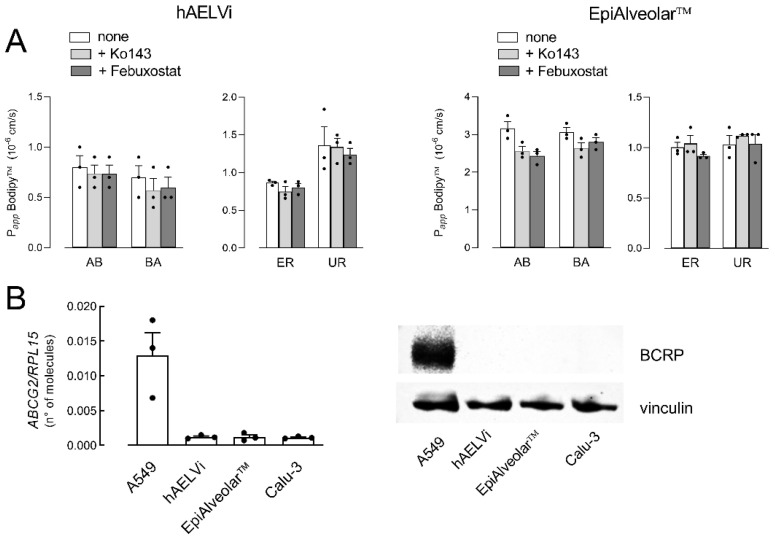
Expression and activity of *ABCG2*/BCRP. Panel (**A**): The apical-to-basolateral (AB) and basolateral-to-apical (BA) fluxes of 3 µM Bodipy™ FL Prazosin were monitored both in the absence (none) and in the presence of 20 µM Ko143 and 50 µM Febuxostat, as indicated. Data obtained were employed to calculate the efflux ratio (ER) and uptake ratio (UR), as defined in Materials and Methods. Bars are the mean ± SEM of 3 independent determinations (single dots). Panel (**B**): The expression of *ABCG2*/BCRP gene and protein were performed with RT-qPCR and Western blot analyses, respectively, as described in Materials and Methods. Data of mRNA expression are the mean ± SEM of 3 independent determinations (single dots), while a representative Western blot is shown that, repeated twice, gave comparable results.

**Figure 4 biomolecules-12-01260-f004:**
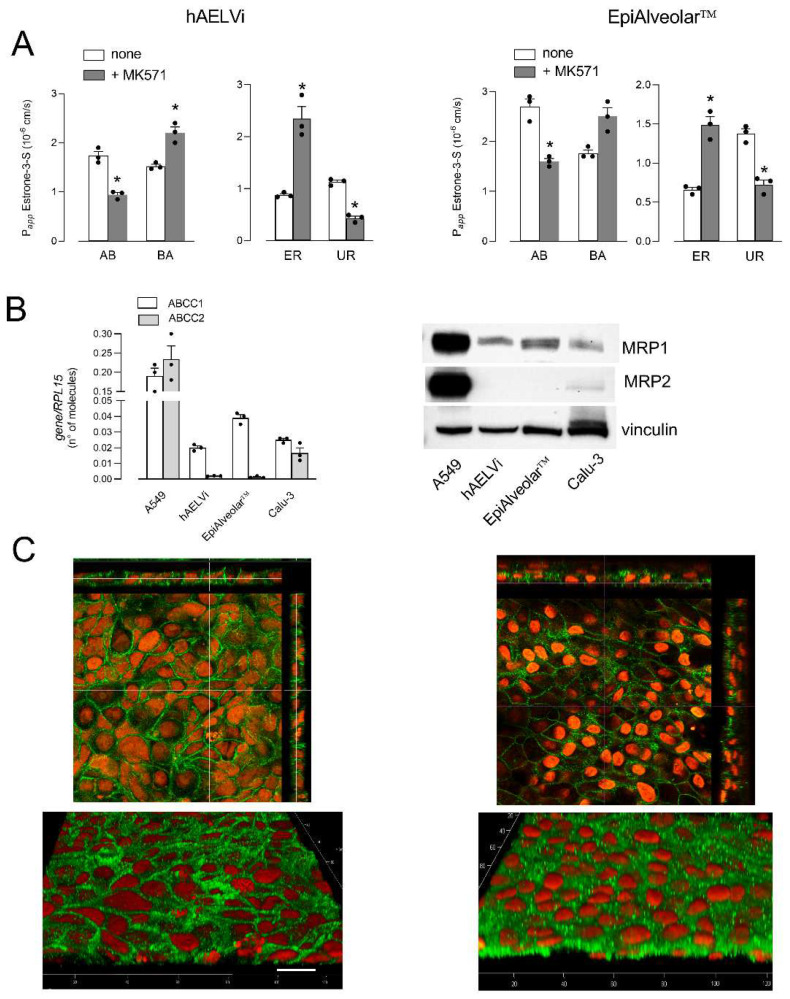
Expression and activity of *ABCCs/*MRPs. Panel (**A**): The apical-to-basolateral (AB) and basolateral-to-apical (BA) fluxes of 0.5 µM ^3^H-estrone-3-sulfate were monitored both in the absence (none) and in the presence of 50 µM MK571, as indicated. Data obtained were employed to calculate the efflux ratio (ER) and uptake ratio (UR), as defined in Materials and Methods. Bars are the mean ± SEM of 3 independent determinations (single dots). * *p* < 0.05 vs. none with a two-tailed Student’s *t*-test. Panel (**B**): The expression of *ABCC1/*MRP1 and *ABCC2*/MRP2 genes and proteins were performed with RT-qPCR or and Western blot analyses, respectively, as described in Materials and Methods. Data of mRNA expression are the mean ± SEM of 3 independent determinations (single dots), while representative Western blots are shown that, repeated twice, gave comparable results. Panel (**C**): Immunolocalization of MRP1 in hAELVi and EpiAlveolar™. Confocal laser scanning microscopy of monolayers immunolabeled for MRP1 (green) is shown. Nuclei were stained with propidium iodide (red). Top, single XY scans of hAELVi (left) and EpiAlveolar™ (right) are shown; XZ sections of the planes are also shown in the rectangle images. Bottom, tridimensional reconstruction of acquired horizontal sections. Representative images from three independent experiments are shown. Bar = 20 µm.

## Data Availability

The data presented in this study are contained within the article.

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
