# Peer review of "Expression and Function of ABC Transporters in Human Alveolar Epithelial Cells"

_biomolecules, 2022, doi:10.3390/biom12091260_

Round 1

Reviewer 1 Report

The manuscript describes the expression analysis of three ABC transporters in the EpiAlveolar model of human alveoli. Although the main question of the study remains very simple, the investigation is well conducted, overall.

I’m wondering why only those three ABC transporters were evaluated? It has been reported that other ABC transporters (for example ABCA1 and G1) are expressed in alveolar tissue and play very important physiological role. Wouldn’t be interesting to have a look on that? Is it possible or maybe it exist the RNA/Protein expression profile from those cell lines?

How PSC 833 is a specific MDR1 inhibitor since rhodamine 123 might be transported by other ABCs? The same goes for other inhibitors used in the study. It is known that often many inhibitors are not very specific and thus may have a pleiotropic effects.

I would suggest also to include 2D images of MRP1 expression in both cells on a top of 3D reconstruction.

Author Response

We would like to thank the Reviewer for the criticisms he/she raised, that significantly helped improving our manuscript; according to his/her observations, we revised the manuscript as specified below.

Q1. I’m wondering why only those three ABC transporters were evaluated? It has been reported that other ABC transporters (for example ABCA1 and G1) are expressed in alveolar tissue and play very important physiological role. Wouldn’t be interesting to have a look on that? Is it possible or maybe it exist the RNA/Protein expression profile from those cell lines?

A1. We agree with the Reviewer that  ABC transporters other than MDR1, MRP1 and BCRP are expressed in the alveoli and are of peculiar relevance for the homeostasis of the tissue. However, in our contribution, we intended to focus on those known to be the most significant determinants in the transport of exogenous pharmacological compounds, and, particularly, of inhaled drugs; that’s why we chose MDR1, MRP1 and BCRP as targets for our study. This issue is now better clarified in the Introduction, with the addition of more references. The suggestion of the Reviewer is, anyway, of undoubted interest; the expression/activity of other ABC family members in alveolar cells, such as  ABCA1 and ABCG1, very important for cholesterol efflux,  could be  addressed in the context of  the study of surfactant metabolism.

Q2. How PSC 833 is a specific MDR1 inhibitor since rhodamine 123 might be transported by other ABCs? The same goes for other inhibitors used in the study. It is known that often many inhibitors are not very specific and thus may have a pleiotropic effects.

A2. The Reviewer is right when stating that “rhodamine 123 might be transported by other ABCs”; we are aware, indeed, that most of the inhibitors, as well as of the substrates, available are not specific for a single transporter. In our hands, however, the lack of a directionality for rhodamine fluxes excludes the presence/activity of efflux pumps, as further supported by the ineffectiveness of PSC833. In any case, we think that the definitive proof of the absence of MDR1 in our cell model comes from the molecular evidences obtained at mRNA and, especially, at protein level.

Q3. I would suggest also to include 2D images of MRP1 expression in both cells on a top of 3D reconstruction.

A3. As suggested by the Reviewer, 2D images of MRP1 protein have been added to Figure 4.

Reviewer 2 Report

The manuscript entitled, “Expression and function of ABC transporters in human alveolar epithelial cells” by Visigalli et al., briefly discusses the use of an in vitro model called the “EpiAlveolarTM” to study the expression and activity of ABC transporters in Human Alveolar Epithelial Lentivirus-immortalized cells (hAELVi) and normal alveolar epithelial cells. EpiAlveolarTM is an in vitro organotypic 3D model of the human alveolar tissue which the authors have developed and validated in previous studies. Overall, the study design is aptly discussed and the manuscript well-written. I do have some observations which has been listed below

Major Comments:

1.    Line 103-105: Why a trans-epithelial electrical resistance (TEER) value > 500 Ω/cm2 is considered as an index of tight monolayers. Kindly add a reference to this statement or state the reasons for this acceptance criteria?

2.    Line 114-115: Authors state Mannitol permeability coefficient (Papp) values < 5 X 10-7 cm/s were accepted as an index of tight monolayers. Kindly add a reference to this statement or state the reasons for this acceptance criteria?

3.    Figure 1: Papp values for mannitol and propranolol are > 5 x10-7. So based on the acceptance criteria, both hAELVi and EpiAlveolarTM have weak monolayers. But the TEER values suggest these layers are tight. Why this discrepancy? Briefly discuss

4.    Figure 1 graphs need to be re-labeled as A, B, and C; and the figure caption modified accordingly.

5.    Line 202-203: A slight difference was observed…What do the authors mean by this statement?  Was there a statistical difference or not for Papp values for mannitol and propranolol permeability? How many replicates were considered? Authors kindly make a statistically relevant statement.

6.    What was the reason for choosing PSC833 as the P-gp inhibitor? Why only one P-gp inhibitor was studied for the flux studies?  Authors should try to include atleast data from 3 P-gp inhibitors for validation

7.    Similarly, only one MRP1 inhibitor was tested for flux studies. Authors should validate the results with atleast three MRP1 inhibitor.

Minor Comments

1.    Fig 2B: Discontinuous Y axis could be used to better visualize values for A549, hAELVi and EpiAlveolarTM

2.    Kindly add a statement on how the p values were computed in all the figure captions.

3.    Line 189-190: Address grammatical error. It should be “purchased”.

Author Response

We would like to first thank the Reviewer for his/her collaborative criticisms; here below he/she can find the answers to his/her questions.

Major comments:

Q1. Line 103-105: Why a trans-epithelial electrical resistance (TEER) value > 500 Ω/cm2 is considered as an index of tight monolayers. Kindly add a reference to this statement or state the reasons for this acceptance criteria?

A1. References sustaining our criteria for acceptance of monolayer tightness have been added, as required.

Q2. Line 114-115: Authors state Mannitol permeability coefficient (Papp) values < 5 X 10-7 cm/s were accepted as an index of tight monolayers. Kindly add a reference to this statement or state the reasons for this acceptance criteria?

A2. The value we indicated for mannitol Papp has an index of tight monolayers was wrong; the right value is < 5 X 10-6 cm/s, as indicated by the literature evidences that we now added. We apologize for the mistake.

Q3. Figure 1: Papp values for mannitol and propranolol are > 5 x10-7. So based on the acceptance criteria, both hAELVi and EpiAlveolarTM have weak monolayers. But the TEER values suggest these layers are tight. Why this discrepancy? Briefly discuss.

A3. As specified in A2, we made a mistake when writing the Papp value to be considered for monolayer integrity; we now corrected it.

Q4. Figure 1 graphs need to be re-labeled as A, B, and C; and the figure caption modified accordingly.

A4. The labels A, B and C have been added to Figure 1, as required by the Reviewer.

Q5. Line 202-203: A slight difference was observed…What do the authors mean by this statement?  Was there a statistical difference or not for Papp values for mannitol and propranolol permeability? How many replicates were considered? Authors kindly make a statistically relevant statement.

A5. We agree with the Reviewer that the statement was misleading, since no statistical significance was reached (three independent replicates were employed, as specified in the Figure caption). The text has been modified accordingly.

Q6. What was the reason for choosing PSC833 as the P-gp inhibitor? Why only one P-gp inhibitor was studied for the flux studies?  Authors should try to include at least data from 3 P-gp inhibitors for validation

Similarly, only one MRP1 inhibitor was tested for flux studies. Authors should validate the results with at least three MRP1 inhibitor.

A6. It is well recognized that most of the inhibitors, as well as of the substrates, available are not specific for a single transporter and may have pleiotropic effects, leading to confusing findings. For this reason, we think that the definitive proof of the absence/presence of a transporter better comes from the molecular evidences obtained at mRNA and, especially, at protein level, rather than from the use of more than one compound.

Minor comments:

Q7. Fig 2B: Discontinuous Y axis could be used to better visualize values for A549, hAELVi and EpiAlveolarTM

A7. Y axis in Figure 2B has been interrupted, as suggested by the Reviewer.

Q8. Kindly add a statement on how the p values were computed in all the figure captions.

A8. P values were calculated with a two tailed Student's t-test, as indicated in Material and Methods; the issue is now specified in all captions. A sentence has been also added to specify that only P  values < 0.05 were considered as significant.

Q9. Line 189-190: Address grammatical error. It should be “purchased”.

A9. We apologize for the mistake; grammar has been revised throughout the manuscript.

Reviewer 3 Report

The manuscript characterizes the expression of the three primary ABC transporters, ABCB1 (formerly MDR1), ABCG2 (formerly BCRP), and ABCC1(formerly MRP1), in these model systems. However, the abundance and activity of the many other ABC transporters (altogether 48 in humans) are not studied in the current work. The authors use a transport system and inhibitors to assess the function of ABC transporters in transcellular transport experiments. With a careful selection of a few substrates and inhibitors relatively specific to the three ABC transporter in question, the authors can clearly discriminate the function and relative importance of these three transporters in these alveolar models. However, although these three transporters are generally the most significant determinants of exogenous pharmacological compounds, definitely that of chemotherapeutic drugs, by playing an important role in multidrug resistance, there are many other ABC transporters with overlapping substrate specificity and inhibitor profiles matching that of these three transporters. Many of them have a significant physiological and pharmacological relevance as well.

The major finding of the work is that ABCC1 is expressed at the basolateral plasma membrane in the EpiAlveolar model, and the authors find a predominantly unidirectional flux of 3H estrone-3 sulfate, which is sensitive to MK571 inhibition. This would fit an ABCC1-dependent estrone-3 sulfate transport.

However, there is no solid evidence that this MK571 sensitive 3H estrone-3 sulfate flux is really mediated by ABCC1. A few other C-type ABC transporters (ABCC2, ABCC3) can also mediate 3H estrone-3 sulfate transport; generally, C-type ABC transporters are almost all inhibited by MK571, with a similar inhibitory constant. Therefore, without further detailed analysis, this 3H estrone-3 sulfate flux cannot be interpreted as mediated by ABCC1. Therefore, this conclusion needs to be changed throughout the entire manuscript, and the function needs to be more generally attributed as a "C-type ABC transporter-dependent activity". Alternatively, a more detailed analysis should be performed. The mRNA and protein expression of at least the following ABC transporters should be investigated: ABCC2 and ABCC3. Moreover, silencing or knockdown of these two transporters should be performed to find out which of the three closely related transporters (ABCC1, ABCC2, ABCC3) with overlapping substrate/inhibitor profiles is indeed responsible for the detected MK571 sensitive 3H estrone-3 sulfate flux. In the lack of these detailed investigations, to tone down the overinterpreted message of the manuscript, the more general "C-type ABC transporter-dependent activity" term should be used instead of dedicating this function solely to ABCC1 without any further direct evidence.

Minor comment:

Please use ABCB1, ABCG2 and ABCC1 for the three transporters throughout the manuscript as MDR1/Pg-p, BCRP and MRP1 is the old nomenclature.

Author Response

First, we would like to thank the Reviewer for the attention he/she paid to our manuscript; here below he/she can find the answers to his/her questions.

Major comments:

However, there is no solid evidence that this MK571 sensitive 3H estrone-3 sulfate flux is really mediated by ABCC1. A few other C-type ABC transporters (ABCC2, ABCC3) can also mediate 3H estrone-3 sulfate transport; generally, C-type ABC transporters are almost all inhibited by MK571, with a similar inhibitory constant. Therefore, without further detailed analysis, this 3H estrone-3 sulfate flux cannot be interpreted as mediated by ABCC1. Therefore, this conclusion needs to be changed throughout the entire manuscript, and the function needs to be more generally attributed as a "C-type ABC transporter-dependent activity". Alternatively, a more detailed analysis should be performed. The mRNA and protein expression of at least the following ABC transporters should be investigated: ABCC2 and ABCC3. Moreover, silencing or knockdown of these two transporters should be performed to find out which of the three closely related transporters (ABCC1, ABCC2, ABCC3) with overlapping substrate/inhibitor profiles is indeed responsible for the detected MK571 sensitive 3H estrone-3 sulfate flux. In the lack of these detailed investigations, to tone down the overinterpreted message of the manuscript, the more general "C-type ABC transporter-dependent activity" term should be used instead of dedicating this function solely to ABCC1 without any further direct evidence.

The Reviewer is right when stating that “A few other C-type ABC transporters (ABCC2, ABCC3) can also mediate 3H estrone-3 sulfate transport”; we are aware, indeed, that most of the inhibitors, as well as of the substrates, available are not specific for a single transporter. By following the Reviewer’s suggestions, we added results on the expression of MRP2 and demonstrated that this transporter is not expressed. Anyway, the issue raised by the Reviewer has been now better addressed, and the text in the new version of the manuscript has been revised according to his/her suggestions.

Minor comment:

Please use ABCB1, ABCG2 and ABCC1 for the three transporters throughout the manuscript as MDR1/Pg-p, BCRP and MRP1 is the old nomenclature.

We agree with the Reviewer that MDR1, BCRP and MRP1 correspond to the old nomenclature, but we feel that these names may still help the reader in understanding our findings; only Pg-p has been eliminated throughout the text.

Round 2

Reviewer 1 Report

I'm ok with the changes.

Author Response

Thank you

Reviewer 2 Report

Thank you reviewers for addressing my earlier raised concerns regarding the manuscript. All the comments have been addressed satisfactorily. I wish the authors all the best in their future research.

Author Response

Thank you very much for your suggestions, that greatly helped to improve our contribution

Reviewer 3 Report

I appreciate that the Authors investigated the expression of MRP2 in the model systems, albeit they did not investigate that of other C type ABC transporters. In the revised manuscript the conclusion regarding 3H-estrone-3-sulphate transport and its inhibition by MK571 is modified and is now interpreted as "ABCC-mediated activity", what I accept to be a valid conclusion.

Minor comment: Please indicate the exact name/catalog number of the antibodies used in the work.

Author Response

According to your suggestion, the catalog numbers of the antibodies used have been added. Thank you again for your comments.